# Dynamic Modeling and Analysis of Nonlinear Compound Planetary System

**Tingqiong Cui [1], Yinong Li [1,\*], Chenglin Zan [1] and Yuanchang Chen [2]**

[1] College of Mechanical and Vehicle Engineering, Chongqing University, Chongqing 400030, China; cuitq@cqu.edu.cn (T.C.); zancl@cqu.edu.cn (C.Z.)

[2] Structural Dynamics and Acoustic Systems Laboratory, University of Massachusetts Lowell, One University Avenue, Lowell, MA 01854, USA; y.c.chen1990@gmail.com

\* Correspondence: ynli@cqu.edu.cn; Tel.: +86-130-7543-1806

**Abstract:** In the vehicle composite planetary gear transmission system, nonlinear excitations such as time-varying meshing stiffness, backlash and comprehensive error would lead to large vibration and noise, uneven load distribution, unstable operation and other problems. To address these issues, this work focuses on compound planetary gears and develops the bending-torsion coupling nonlinear dynamic model of the system based on the Lagrange equation. There are internal and external multi-source excitations applied to the system. This model is used to study the bending-torsion coupling meshing deformation relationship of each meshing pair along with the translational and torsional directions. The natural frequencies and vibration modal characteristics of the system are extracted from the model, and the influence of rotational inertia and meshing stiffness on the inherent characteristics of the system are studied. The coupling vibration characteristics of the system under operating condition are analyzed in terms of the inherent characteristics and time–frequency characteristics of the system. The simulation results exhibit that the planetary gear system has three modes. The change in natural frequency trajectory has two phenomena: modal transition and trajectory intersection. The main frequencies include engine rotating frequency, meshing frequency and its double frequency, and the rotation frequency and harmonic frequency of the engine have a great influence on the vibration response of the system. Finally, the virtual prototype of the composite planetary system is used to verify the accuracy of the established model from speed, inherent characteristics, meshing force and frequency composition.

**Keywords:** compound planetary gear; non-linear excitation; composite error; meshing clearance; time-frequency characteristics; meshing force

## 1. Introduction

The planetary gear transmission system has, as its characteristics, a large transmission ratio, stable transmission, strong carrying capacity, high unit volume, mass power density, coaxial input and output and strong impact resistance and power shunt, which is widely used in various industrial fields. With the change in the number and phase of the planetary gear, as well as the diversification of the engagement with the gear ring sun gear, the design of the planetary gear system becomes complex and diverse. There are some nonlinear excitations in the gear transmission system, such as time-varying meshing stiffness, backlash and comprehensive transmission error, which cause the vibration and noise of the system, and directly affect the reliability and safety of the system.

Many scholars have made an in-depth study of the vibration characteristics of planetary systems. Ambarisha et al. [1] applied the lumped parameter and finite element model to investigate the complex and nonlinear dynamic behavior of spur planetary gears. Sun et al. [2–4] studied the strongly nonlinear dynamic behavior of planetary gear systems with multiple clearances. Kim et al. [5] offered the planetary gear dynamic model, assuming that the position of the contact line is determined by the average angular motion

of the gear. Zhou et al. [6] employed numerical methods to analyze the planetary gear transmission system at high speed and light weight. Liu et al. [7–9] made a study of the planetary gear dynamic model considering time-varying stiffness, gear meshing error and gyroscopic effect. Kahraman et al. [10] established a planetary gear pure torsion dynamics model to obtain the planetary gear's dynamics in which the natural frequency is compared with the dynamic model of the torsion translation planetary gear. Peng et al. [11] first used the meshing phase to study planetary gears with faults. Qiu et al. [12] reviewed the dynamics of the planetary gear transmission system of wind turbines from the aspects of research status, dynamic optimization design and development direction. Xiang et al. [13] discovered the non-linear dynamic characteristics of the system under the change in the planetary gear's supporting stiffness and excitation frequency. Bozca et al. [14] presented a multi-body torsional vibration dynamic model of the rigid-flexible coupling planetary gear system by using the state-space model and considering the influence of box flexibility. Zhang et al. [15] raised a dynamic model of planetary transmission considering the flexibility of the ring gear and identified its inherent characteristics and relevant test verification. Qin et al. [16] advanced the coupling dynamics model of the wind turbine gear transmission system and examined the natural frequency, vibration response, dynamic meshing force and rolling bearing dynamic bearing force of the wind turbine gear transmission system. Dou et al. [17,18] studied the frequency coupling and coupling resonance of the composite planetary transmission system under complex excitations. Wang et al. [19] studied the time–frequency characteristics of planetary gear systems by establishing a flexible multi-body dynamics model. Li et al. [20] inspected the spectral characteristics of vibration signals of the compound planetary transmission system. Zhu et al. [21] explored the dynamic response characteristics of planetary transmission systems by gear modification. Inalpolat et al. [22] investigated the influence of system input speed on the dynamic response amplitude of multi-stage planetary gear systems and verified it through experiments. Chaari et al. [23] utilized the iterative method to explore the dynamic response characteristics of planetary transmission systems under nonlinear factors. Zhang et al. [24] established the dynamic model of planetary gear systems with sliding bearing and flexible structure, and the influence of different radial sliding bearings on load sharing characteristics was researched. Gu et al. [25] proposed a lumped parameter model of original planetary gear considering planetary position error and simulated their effects on quasi-static and dynamic load sharing among planets. Li et al. [26] observed the influence law of each excitation factor from the perspective of bifurcation characteristics, and researched whether the change in backlash would affect whether the system was periodic or chaotic. Ryali et al. [27] proposed a three-dimensional dynamic load distribution model of planetary gear sets. Mo et al. [28] inspected the load-sharing characteristics of flexible support when the sun gear is floating and normal. Sanchez-Espiga et al. [29] proposed a numerical approach for calculating the load distribution of planetary transmission. Iglesias et al. [30] found that the tooth load is much lower than the planetary non-uniform load in the transmission with and without defects.

These scholars have done a large amount of work on the dynamic modeling of single-stage planetary row, nonlinear dynamic analysis and load sharing characteristics. Few people have studied the dynamic characteristics of multi-stage planetary gear system under internal and external nonlinear excitation. However, these nonlinear factors bring about self-excited motion, which has a great influence on the system. In order to study the vibration problem caused by nonlinear excitation in the composite planetary system, this paper establishes the bending-torsion coupling dynamic model of the composite planetary transmission system, considering the external and internal excitations such as engine harmonic excitation, time-varying stiffness, gear time-varying phase, comprehensive error and dynamic backlash. The meshing deformation relationship of meshing pair in translational and torsional directions was analyzed in detail, and the correctness of the model is verified by the virtual prototype from the aspects of rotational speed, inherent characteristics, meshing force and frequency composition. The vibration characteristics of the nonlinear

system are revealed by analyzing the natural frequency, which varies with stiffness and inertial trajectory, as well as the time–frequency characteristics of the system response.

## 2. Dynamic Model for Compound Planetary System

### 2.1. Model Description

The schematic of the compound planetary system is shown in Figure 1, which includes three pairs of planetary chains composed of sun gear $s$, planet carrier $c$, long planet gear $a$, short planet gear $b$, the small gear ring $r_1$ and the big gear ring $r_2$. In addition, the brakes $C_1$ and $C_2$, and the clutch $C_3$, control different gears. The meshing relations in the system are equivalently simplified into a mass-spring-damping model, as shown in Figure 1. In this paper, the brake $C_1$ is engaged, and the brake $C_2$ and the clutch $C_3$ are disengaged. The small ring gear $r_1$ sends power to the long planetary gear $a$, while $a$ transmits the power to the sun gear s and the short planetary gear $b$; $b$ transfers the power to the large ring gear $r_2$ and the planet carrier $c$; finally, the power is output from both ends of the planet carrier.

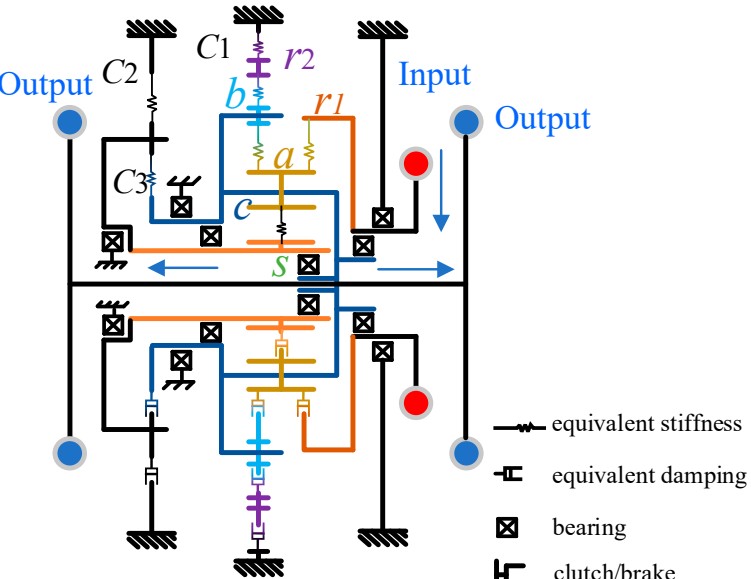

**Figure 1.** Schematic of the compound planetary transmission system.

To describe the meshing relationship between planetary gear systems more accurately, three coordinate systems are used in this paper: the first is the absolute coordinate system $X\_O\_Y$, which is used to describe the vibration displacement of all central rotating parts; the second is the involved coordinate system $X_i\_O_i\_Y_i$, which describes the translational vibration displacement and which rotates synchronously with the planetary shelf; the third is the relative coordinate system $x_{aj}\_O_{aj}\_y_{aj}$ of the jth planetary wheel in the planetary row of a or b, whose origin is located in the center of each planetary wheel, and its coordinate axis is orthogonally along the radial and circumferential directions of the planetary carrier. In the dynamic model of the composite planetary gear system established in this paper, the vibration displacement of planetary gear is based on their respective relative coordinate systems $x_{aj}\_O_{aj}\_y_{aj}$. The bending–torsion coupling mechanical model of the composite planetary gear system is shown in Figure 2. $k_{msaj}, k_{mraj}, k_{mrbj}$ and $k_{mabj}$ are the meshing stiffness of the meshing pair in the torsional direction; $\theta_c, \theta_{r1}, \theta_{r2}$, and $\theta_s$, respectively, represent the torsional displacement relative to the fixed reference frame $X\_O\_Y$; $\theta_{aj}$ and $\theta_{bj}$ indicate the relative torsional displacement of the planet wheel $a_j$ and the planet wheel $b_j$ in the relative coordinate system $x_{aj}\_O_{aj}\_y_{aj}, x_{bj}\_O_{bj}\_y_{bj}$, where $j = 1, 2, 3$; $k_{bpx}, k_{bqx}, k_{bpy}$, and $k_{bqy}$ express the supporting stiffness of p and $q$ in the $x$ direction and $y$ direction in their respective coordinate systems, where $p = s, c, r1$ and $r2$, respectively, mean the sun wheel, planet rack, small ring gear and large ring gear in the

composite planetary gear system; $q = aj, bj$ represent the long planet wheel and the short planet wheel in the composite planetary gear.

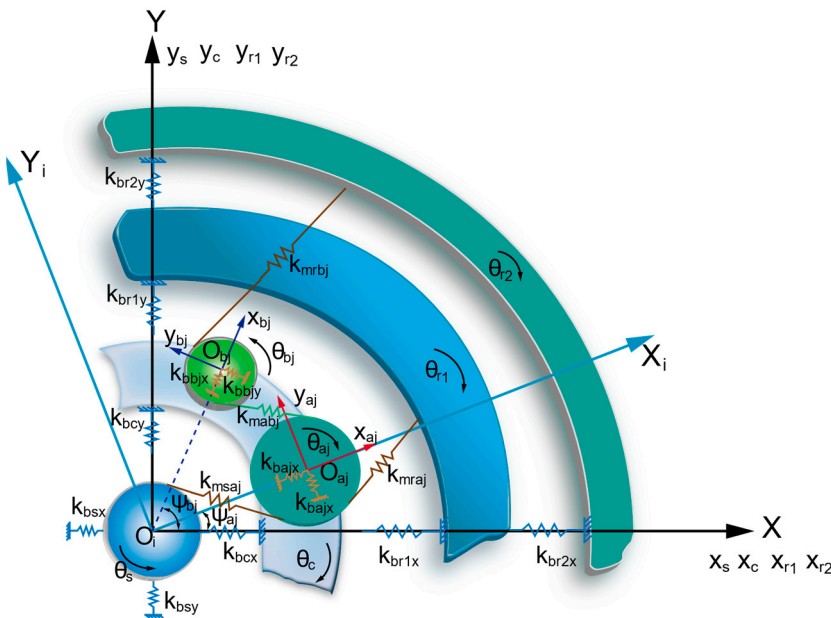

**Figure 2.** Bending–torsional coupling mechanical model of composite planetary gear system.

### 2.2. The Deformation of Meshing Planetary Gears

The expression of meshing deformation of a pair of gear teeth is related to the meshing model, coordinate selection, positive rotation direction selection and meshing line selection. Meshing deformation is defined along the tangential direction of the base circle between meshing gears. There are four meshing relationships in Figure 3 in this paper; the meshing deformation relationship between planetary wheels is selected for research and the meshing compression is positive in the direction of deformation.

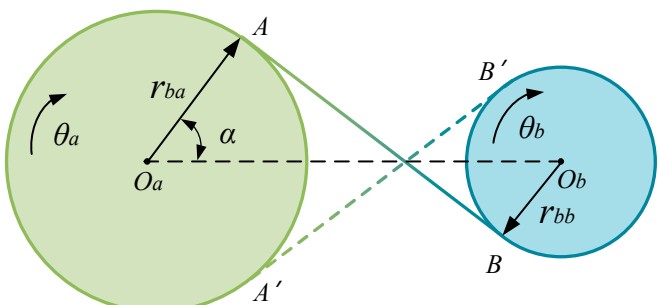

**Figure 3.** Torsional meshing deformation relationship between the planetary gear $a_j$ and the planetary gear $b_j$.

There are two different mesh lines, $AB$ and $A'B'$, in Figure 3, though their meshing deformation calculation method is the same. This paper selects the $AB$ as the mesh line. $r_{ba}$ and $r_{bb}$ are the base circle radius of the two gears. The mesh deformation caused by pure torsion of the gear can be described by the length variation between $A$ and $B$ points, which can be expressed as Equation (1):

$$\delta_{ab} = r_{ba}\theta_a + r_{bb}\theta_b \tag{1}$$

The projection relation of the translational displacement of the planetary wheel $a_j$ on the meshing line is shown in Figure 4a. The absolute translational displacement of

the planetary rack $L_c$ is the vector sum of translational displacements $x_c$ and $y_c$ in the fixed coordinate, which is also the involvable displacement of the planetary wheel; $x_c$ and $y_c$ are projected to the orthogonal directions of $x_{aj}$ and $y_{aj}$; the relative displacement $L'_a$ of the planet wheel $a_j$ is the vector sum of the translational displacements $x_{aj}$ and $y_{aj}$ in the relative coordinate $x_{aj}O_{aj}y_{aj}$; the absolute translational displacements $L_a$ can be regarded as the vector sum of its relative translational displacements $L'_a$ and the involved translational displacements $L_c$; finally, the total deformation of $x_{aj}$ and $y_{aj}$ in two orthogonal directions is projected along the meshing line EF to obtain the change in the planetary wheel $a_j$ on the meshing line EF induced by the translation displacement $\sigma^b_{ab}$, as shown in Equation (8). Similar to the planetary gear $a_j$, the projection relationship of the planetary gear $b_j$'s translation displacement on the meshing line is shown in Figure 4b.

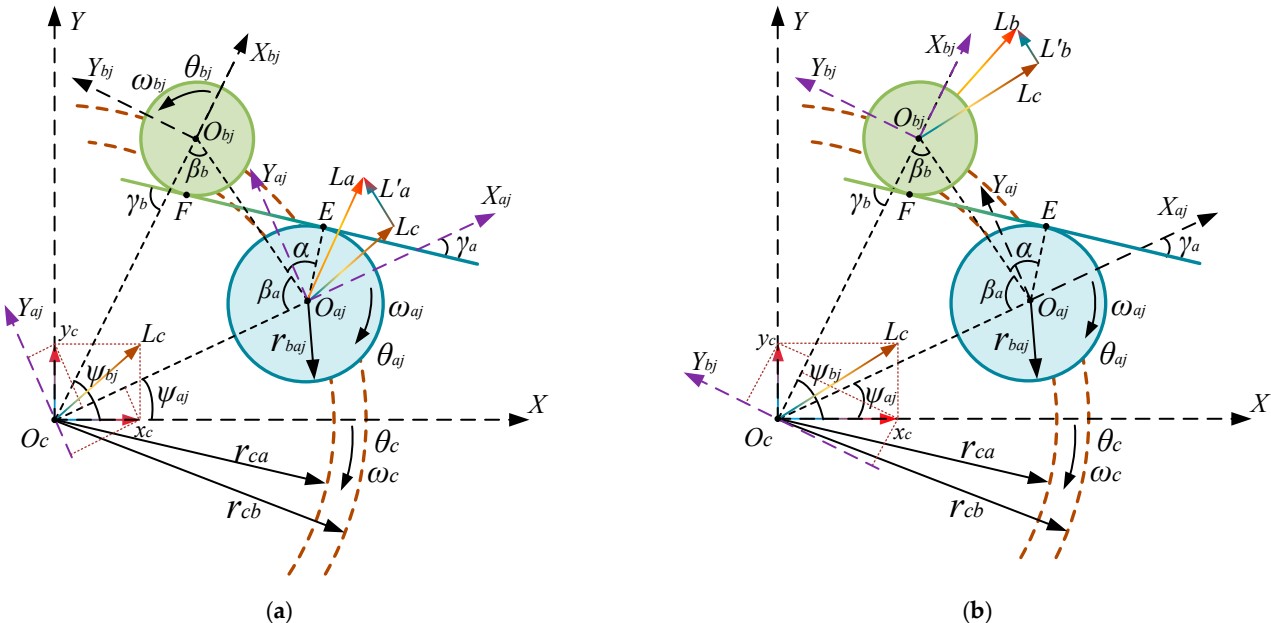

**Figure 4.** Translational meshing deformation relationship between the planetary gears $a_j$ and $b_j$: (**a**) Projection relation of $a_j$ translation displacement; (**b**) Projection relation of $b_j$ translation displacement.

In Figure 4, $r_{ca}$ and $r_{cb}$, respectively, express the distance between the theoretical rotation center of the planet wheel $a_j$ and $b_j$ and the rotation center of the planet rack. $\psi_{aj}$ and $\psi_{bj}$ represent the position angle of the planet wheels $a_j$ and $b_j$ in the fixed coordinate system $XOY$, which are calculated as shown in Equations (2) and (3). $\beta_a$ and $\beta_b$ indicate the angle between the centerline of the planetary wheel $a_j$ and $b_j$, meshing with each other and their respective coordinate axes; their relationship is shown in Equations (4) and (5). $\gamma_a$ and $\gamma_b$ indicate the angle between the meshing line of the $a_j$ and $b_j$ intermeshing planetary wheels and their respective coordinate axes $X_{aj}$ and $X_{bj}$, which are calculated as shown in Equations (6) and (7); the other symbols have the same meaning as previously stated.

$$\psi_{aj} = \omega_c t + 2\pi(j-1)/n + \varphi_a \tag{2}$$

$$\psi_{bj} = \omega_c t + 2\pi(j-1)/n + \varphi_b \tag{3}$$

in which $n$ is the number of planetary gears; $\varphi_a$ and $\varphi_b$ denote the initial circumferential installation angle of planetary gear $a_j$ and $b_j$.

$$\beta_a + \beta_b = \psi_{bj} - \psi_{aj} \tag{4}$$

$$\frac{\sin\beta_a}{r_{cb}} = \frac{\sin\beta_b}{r_{ca}} \tag{5}$$

$$\gamma_a = \beta_a + \alpha - \pi/2 \tag{6}$$

$$\gamma_b = \pi/2 - (\beta_b - \alpha) \tag{7}$$

The total deformation $\sigma_{ab}^a$ of planetary gear $a_j$ on the meshing line EF is:

$$\sigma_{ab}^a = -\left(y_c sin\psi_{aj} + x_c cos\psi_{aj} + x_{aj}\right)cos\gamma_a + \left(y_c cos\psi_{aj} - x_c sin\psi_{aj} + y_{aj}\right)sin\gamma_a \tag{8}$$

The total deformation $\sigma_{ab}^a$ of planetary gear $b_j$ on the meshing line EF is:

$$\sigma_{ab}^b = \left(y_c sin\psi_{bj} + x_c cos\psi_{bj} + x_{bj}\right)cos\gamma_b - \left(y_c cos\psi_{bj} - x_c sin\psi_{bj} + y_{bj}\right)sin\gamma_b \tag{9}$$

The total meshing deformation $\sigma_{ab}$ caused by the translation displacement in the meshing line direction of planetary gear $a_j$ and planetary gear $b_j$ is:

$$\sigma_{ab} = \sigma_{ab}^a + \sigma_{ab}^b \tag{10}$$

Similarly, in the direction of the outer meshing line between the sun wheel and the planetary gear, the meshing deformation $\delta_{sa}$ and the total meshing deformation in external meshing line $\sigma_a^s$ are shown in Equations (11) and (13). In the direction of the internal meshing line between the gear ring and the planetary gear, the meshing deformation $\delta_{ra}$ and the total meshing deformation in external meshing line $\delta_a^r$ are shown in Equations (14) and (16).

$$\delta_{sa} = \lambda_{sa}\left(r_{bs}(\theta_s - \theta_c) + r_{ba}\theta_a\right) = \sigma_s + \sigma_a^s \tag{11}$$

$$\sigma_s = x_s sin\left(\psi_{aj} + \alpha\right) - y_s cos\left(\psi_{aj} + \alpha\right) \tag{12}$$

$$\sigma_a^s = -x_c sin\left(\psi_{aj} + \alpha\right) + y_c cos\left(\psi_{aj} + \alpha\right) - x_{aj}sin\alpha + y_{aj}cos\alpha \tag{13}$$

$$\delta_{ra} = \lambda_{ra}\left(r_{ba}\theta_a + r_{br}(\theta_r - \theta_c)\right) = \sigma_r + \sigma_a^r \tag{14}$$

$$\sigma_r = -x_r sin\left(\psi_{aj} - \alpha\right) + y_r cos\left(\psi_{aj} - \alpha\right) \tag{15}$$

$$\sigma_a^r = x_c sin\left(\psi_{aj} - \alpha\right) - y_c cos\left(\psi_{aj} - \alpha\right) - x_{aj}sin\alpha - y_{aj}cos\alpha \tag{16}$$

### 2.3. Internal Nonlinear Excitation

The internal nonlinear excitation would be caused by the comprehensive error, the dynamic backlash and the time-varying stiffness.

The eccentricity error (EE) between the sun and the planets is:

$$\begin{cases} e_{sa}^i = E_s sin(-\omega_s t - \varepsilon_s + \psi_{sa}^i + \alpha) \\ e_{as}^i = -E_a^i sin(-\omega_a^i t - \varepsilon_a^i + \alpha) \end{cases} \tag{17}$$

The EE between gear ring $r1$ and planetary gear $a$ is:

$$\begin{cases} e_{ra}^i = -E_r sin(\omega_r t + \varepsilon_r - \psi_{ra}^i + \alpha) \\ e_{ar}^i = E_a^i sin(\omega_a^i t + \varepsilon_a^i + \alpha) \end{cases} \tag{18}$$

The EE between gear ring $r2$ and planetary gear b is:

$$\begin{cases} e_{rb}^i = -E_r sin(-\omega_r t - \varepsilon_r + \psi_{rb}^i + \alpha) \\ e_{br}^i = E_b^i sin(-\omega_b^i t - \varepsilon_b^i + \alpha) \end{cases} \tag{19}$$

The EE between planetary wheel a and planetary wheel b is:

$$\begin{cases} e_{ab}^i = -E_a^i sin(\omega_a^i t + \varepsilon_a^i + \alpha + \gamma_a^i) \\ e_{ba}^i = E_b^i sin(\omega_b^i t + \varepsilon_b^i + \alpha - \gamma_b^i) \end{cases} \tag{20}$$

The EE caused by planetary carrier to internal and external meshing pairs is:

$$
\begin{cases}
e^i_{c\_sa} = E_c sin(-\omega_c t + \psi^i_j + \alpha) \\
e^i_{c\_ra} = E_c sin(\omega_c t - \psi^i_j + \alpha) \\
e^i_{c\_ab} = E_c sin(-\omega_c t + \psi^i_j + \alpha) \\
e^i_{c\_rb} = E_c sin(\omega_c t - \psi^i_j + \alpha)
\end{cases}
\tag{21}
$$

where $E_l$ and $E^i_q$ represent the amplitude of manufacturing eccentricity error, $\varepsilon_l$ and $\varepsilon^i_q$ stand for the initial phase of eccentricity error corresponding, $e^i_{lq}$, $e^i_{ql}$ and $e^i_{qq}$ are the eccentricity error of sun gear, gear ring and planetary gear, $l = s, r$ and $c$ are, respectively, sun gear, gear ring and planetary carrier; $q = a, b$ are the two-stage planetary gear; and, $\gamma^i_b$ are the angle between the theoretical installation position of planetary wheel $1pi$ in row a and $2pi$ in row b; $\psi^i_j$ is the position angle between the central component j and the ith planetary wheel.

The assembly eccentricity error (AEE) between the sun gear and planetary wheel a is:

$$
\begin{cases}
a^i_{sa} = A_s sin(-\varphi_s + \psi^i_{sa} + \alpha) \\
a^i_{as} = -A^i_a sin(-\varphi^i_a + \alpha)
\end{cases}
\tag{22}
$$

The AEE between gear ring 1 and planetary gear a is:

$$
\begin{cases}
a^i_{ra} = -A_r sin(-\varphi_r - \psi^i_{ra} + \alpha) \\
a^i_{ar} = A^i_a sin(-\varphi^i_a + \alpha)
\end{cases}
\tag{23}
$$

The AEE between gear ring 2 and planetary wheel b is:

$$
\begin{cases}
a^i_{rb} = -A_r sin(-\varphi_r + \psi^i_{rb} + \alpha) \\
a^i_{br} = A^i_b sin(-\varphi^i_b + \alpha)
\end{cases}
\tag{24}
$$

The AEE caused by planetary carrier to internal and external meshing pairs is:

$$
\begin{cases}
a^i_{c\_sa} = A_c sin(-\varphi_c + \psi^i_j + \alpha) \\
a^i_{c\_ra} = A_c sin(\varphi_c - \psi^i_j + \alpha) \\
a^i_{c\_ab} = A_c sin(-\varphi_c + \psi^i_j + \alpha) \\
a^i_{c\_rb} = A_c sin(\varphi_c - \psi^i_j + \alpha)
\end{cases}
\tag{25}
$$

The AEE between planetary wheel a and planetary wheel b is:

$$
\begin{cases}
a^i_{ab} = -A^i_a sin(\varphi^i_a + \alpha + \gamma^i_a) \\
a^i_{ba} = A^i_b sin(\varphi^i_b + \alpha - \gamma^i_b)
\end{cases}
\tag{26}
$$

where $A_l$ and $A^i_q$ stand for the assembly eccentricity error of amplitude, $\varphi_l$ and $\varphi^i_q$ represent the eccentric error of the initial phase, $a^i_{lq}$, $a^i_{ql}$ and $a^i_{qq}$ are, respectively, projection to the corresponding meshing line wheel, gear ring, the sun and the planets round eccentric error; $l = s, r$ and $c$ are, respectively, sun gear, gear ring and planetary carrier; $q = a, b$ are the two-stage planetary gear.

The tooth profile equivalent meshing error is expressed as a harmonic function, and its excitation frequency is the meshing frequency of the tooth pair, as shown in Equation (27):

$$
\begin{cases}
b^i_{sa} = B^i_{sa} sin(\omega_{msa} t + 2\pi\gamma^i_{sa}) \\
b^i_{ra} = B^i_{ra} sin(\omega_{mra} t + 2\pi\gamma^i_{ra}) \\
b^i_{rb} = B^i_{rb} sin(\omega_{mrb} t + 2\pi\gamma^i_{rb}) \\
b^i_{ab} = B^i_{ab} sin(\omega_{mab} t + 2\pi\gamma^i_{ab})
\end{cases}
\tag{27}
$$

Here, $b_j^i$ is the tooth profile error along the direction of the meshing line, $B_j^i$ is the manufacturing error amplitude, $\omega_{mj}$ is the meshing frequency, $\gamma_j^i$ is the initial phase, $i = 1, 2, 3$ when $j$ equals $sa, ra, rb, ab$, which are, respectively, sun and planetary wheel $p_a^i$, small gear and planetary wheel $p_a^i$, large gear and planetary wheel $p_b^i$ and planetary wheel $p_a^i$.

Based on the mathematical model of the above, the comprehensive error along the engagement line in the composite planetary system is shown in Equation (28):

$$\begin{cases} e_{sa}^i = e_{sa}^i + a_{sa}^i + e_{c\_sa}^i + a_{c\_sa}^i + e_{ab}^i + a_{ab}^i + b_{sa}^i \\ e_{r1a}^i = e_{ra}^i + a_{ra}^i + e_{c\_ra}^i + a_{c\_ra}^i + e_{ba}^i + a_{ba}^i + b_{ra}^i \\ e_{r2b}^i = e_{rb}^i + a_{rb}^i + e_{c\_rb}^i + a_{c\_rb}^i + e_{ba}^i + a_{ba}^i + b_{rb}^i \end{cases} \tag{28}$$

In summary, when considering the bending–torsion coupling mechanical model of the composite planetary gear system, the total deformation on the meshing line is the superposition of the torsional vibration displacement deformation of each gear meshing pair, the deformation induced by the translational vibration displacement and the comprehensive meshing error.

The total meshing deformation (TMD) of the solar wheel and the planetary wheel $a_j$ is:

$$\varepsilon_{saj} = \delta_{saj} + \sigma_{saj} + e_{sa}^{aj} \tag{29}$$

The TMD between the pinion ring and the $a_j$ the planetary gear is:

$$\varepsilon_{raj} = \delta_{raj} + \sigma_{raj} + e_{ra}^{aj} \tag{30}$$

The TMD between the large gear ring and the $b_j$ the planetary gear is:

$$\varepsilon_{rbj} = \delta_{rbj} + \sigma_{rbj} + e_{rb}^{bj} \tag{31}$$

The TMD $\varepsilon_{abj}$ of the $a_j$ planetary gear and the $b_j$ planetary gear is:

$$\varepsilon_{abj} = \delta_{abj} + \sigma_{abj} + e_{ab}^{j} \tag{32}$$

The nonlinear meshing forces of each meshing gear in the x and y directions on the meshing line and in the respective coordinate systems.

$$\begin{cases} F_{mpj} = k_{mpj}(t)f\left(\varepsilon_{pj}, b\right) + c_m \dot{\varepsilon}_{pj} \\ F_{mpjx} = F_{mpj}\sin\left(\psi_{pj} + \alpha\right) \qquad (j = 1, 2, 3) \\ F_{mpjy} = F_{mpj}\cos\left(\psi_{pj} + \alpha\right) \end{cases} \tag{33}$$

when $p$ equals $sa, r$ and $rb$ are, respectively, sun and planetary wheel, small gear and planetary wheel $a_j$ and large gear and planetary wheel $b_j$.

The nonlinear meshing force of the planetary wheel $a_j$ and planetary wheel $b_j$ in the direction is:

$$\begin{cases} F_{mabj} = k_{mabj}(t)f\left(\varepsilon_{abj}, b\right) + c_m \dot{\varepsilon}_{abj} \\ F_{mabjx} = F_{mabj}\sin\alpha \qquad (j = 1, 2, 3) \\ F_{mabjy} = F_{mabj}\cos\alpha \end{cases} \tag{34}$$

Bearing support in the translation direction is considered in the composite planetary gear system. Each component is subjected to bearing support reaction in x and y directions.

$$\begin{cases} F_{bqx} = k_{bqx}x_q + c_{bqx}\dot{x}_q \\ F_{bqy} = k_{bqy}y_q + c_{bpy}\dot{y}_q \end{cases} \tag{35}$$

when $q$ equals $s, r1, r2, a_j$ and $c$ are, respectively, sun gear, small gear, large gear, planetary wheel $a_j$, planetary wheel $b_j$ and planetary frames.

Based on the above deformation relationship and elastic mechanics theory, considering the time-varying stiffness $k_m$ calculated by the Shichuan formula and tooth side clearance function $f(\delta, b)$, the meshing force $F_m$ can be obtained by Equation (36).

$$F_m = k_m(t)f(\delta', b) + c_m \ddot{\delta}' \tag{36}$$

The time-varying stiffness of teeth is:

$$k_m(t) = \frac{F_{load}}{\delta(t)b} \tag{37}$$

in which $F_{load}$ is the normal load on the tooth surface; $b$ is the tooth width; $\delta(t)$ is the total tooth deformation.

Comprehensive deformation $\delta_i$ in the direction of the single tooth meshing line can be expressed as the linear superposition of deformation of rectangular bending part $\delta_{Br}$, local deformation $\delta_G$, shear deformation $\delta_S$ and trapezoidal bending deformation $\delta_{Bt}$.

$$\delta_i = \delta_{Br} + \delta_{Bt} + \delta_S + \delta_G \tag{38}$$

when $i = p, n$, respectively, are the single tooth deformation of driving wheel and driven gear.

The bending deformation of the rectangular part can be expressed as:

$$\delta_{Br} = 12F_N cos^2\omega_x / \left( Ebs_F^3 \left[ h_x h_r (h_x - h_r) + h_r^3 / 3 \right] \right) \tag{39}$$

The bending deformation of trapezoidal part can be expressed as:

$$\delta_{Bt} = 6F_N cos^2\omega_x / (Ebs_F^3) \left[ \frac{h_i - h_x}{h_i - h_r} \left( 4 - \frac{h_i - h_x}{h_i - h_r} \right) - 2\ln\frac{h_i - h_x}{h_i - h_r} - 3 \right] (h_i - h_r)^3 \tag{40}$$

The meshing shear deformation can be expressed as:

$$\delta_S = 2(1 + v)F_N cos^2\omega^2 / Ebs_F \left[ h_r + \left( h_i - h_r \ln\frac{h_i - h_r}{h_i - h_r} \right) \right] \tag{41}$$

The local deformation caused by the inclination of the substrate can be expressed as:

$$\delta_G = 24F_N h_x^2 cos^2\omega^2 / \pi Ebs_F^2 \tag{42}$$

here

$$h_i = (hs_F - h_r s_k) / (s_F - s_k) \tag{43}$$

$$h = \sqrt{r_k^2 - \left(\frac{s_k}{2}\right)^2} - \sqrt{r_r^2 - \left(\frac{s_F}{2}\right)^2} \tag{44}$$

$$h_x = r_x cos(\alpha_x - \omega_x) - \sqrt{r_r^2 - \left(\frac{s_F}{2}\right)^2} \tag{45}$$

when $r_g \leq r_F, z \geq 2\frac{(1-x)}{(1-cos\alpha_0)}$:

$$s_F = 2r_F sin\{-((\pi + 4xtg\alpha_0)/2z) + inv\alpha_0 - inv\alpha_F\} \tag{46}$$

$$\alpha_F = arccos\left(\frac{r_g}{r_F}\right) \tag{47}$$

$$h_r = \sqrt{r_F^2 - \left(\frac{s_F}{2}\right)^2} - \sqrt{r_r^2 - \left(\frac{s_F}{2}\right)^2} \tag{48}$$

when $r_g > r_F$, $z < 2\frac{(1-x)}{(1-cos\alpha_0)}$:

$$s_F = 2r_g sin\{(\pi + 4xtg\alpha_0)/2z + inv\alpha_0\} \tag{49}$$

$$h_r = \sqrt{r_g^2 - \left(\frac{s_F}{2}\right)^2} - \sqrt{r_r^2 - \left(\frac{s_F}{2}\right)^2} \tag{50}$$

where $c$ represents Poisson's ratio; $z$ means number of teeth; $x$ is displacement coefficient; $r_g$ stands for the base circle; $r_r$ denotes the root circle; $r_x$ expresses the pitch circle; $r_k$ is the addendum circle; $r_F$ means radius of effective tooth root circle participating in meshing; $\alpha_0$ is the standard pressure angle; $\alpha_x$ represents meshing angle of the gear.

When a pair of gears mesh, in addition to the deformation caused by each single tooth, there is also the local deformation $\delta_{pv}$ caused by the meshing contact stress $F_n$ of the tooth pair, which can be expressed as

$$\delta_{pv} = 4\left(1 - v^2\right)F_n/\pi Eb \tag{51}$$

Combined with the above formulas, the comprehensive deformation along the meshing direction at the meshing point is

$$\delta(t) = \delta_p + \delta_n + \delta_{pv} \tag{52}$$

### 2.4. External Nonlinear Excitation

This article uses the four-stroke 12-cylinder V-type engine with rated power of 1000 kW and rated speed of 2300 r/min. Under the rated condition, the excitation torque characteristics of the single cylinder and the harmonic analysis of the output torque of the engine are shown in Figure 5a,b. It can be seen that the amplitude characteristics of the output torque of the engine are mainly composed of six-order harmonics, except for the average torque. In the nonlinear dynamic model, the torque can be equivalent to the superposition of the average amplitude and the harmonic function of the six-order harmonics. The expression is shown in Equation (53):

$$M_{in} = M_0 + \mu M_0 sin(2\pi f_e t + \varphi_e) \tag{53}$$

where $M_0$ is the average output torque of the engine, $f_e$ is the torque fluctuation frequency, $\varphi_e$ is the initial phase and $\mu$ is the ratio of the torque amplitude to the torque $M_0$.

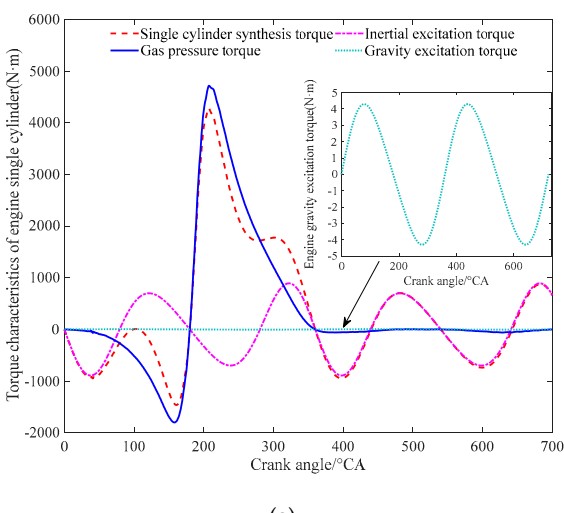

(**a**)

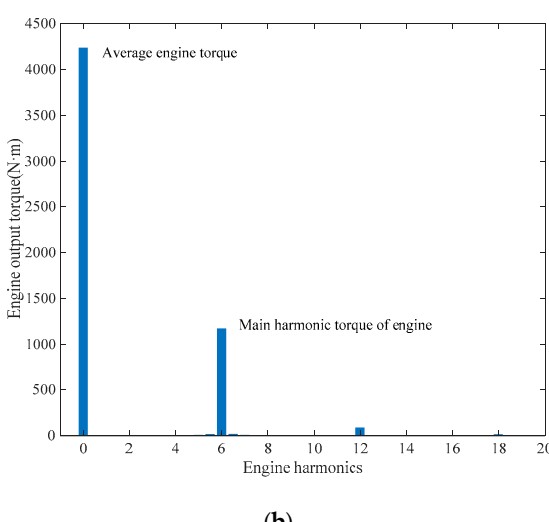

(**b**)

**Figure 5.** Engine harmonic excitation: (**a**) The excitation torque of engine single cylinder; (**b**) Harmonic relation of engine output torque.

### 2.5. Nonlinear Flexural and Torsional Coupling Dynamics Equation of the System

For multi-degree-of-freedom complex dynamical systems, the Lagrange equation is usually used to derive the system vibration differential equation.

The bending–torsional coupling vibration differential equation of sun gear is:

$$
\begin{cases}
m_s \ddot{x}_s + \sum_{j=1}^{3} F_{msajx} + F_{bsx} = 0 \\
m_s \ddot{y}_s - \sum_{j=1}^{3} F_{msajy} + F_{bsy} = 0 \\
J_s \ddot{\theta}_s - \sum_{j=1}^{3} r_{bs} F_{msaj} = 0
\end{cases}
\tag{54}
$$

The bending–torsional coupling vibration differential equation of a small gear ring is:

$$
\begin{cases}
m_{r1} \ddot{x}_{r1} + \sum_{j=1}^{3} F_{mrajx} + F_{br1x} = 0 \\
m_{r1} \ddot{y}_{r1} - \sum_{j=1}^{3} F_{mrajy} + F_{br1y} = 0 \\
J_{r1} \ddot{\theta}_{r1} + \sum_{j=1}^{3} r_{br1} F_{mraj} = T_{in}
\end{cases}
\tag{55}
$$

The bending–torsional coupling vibration differential equation of the long planetary wheel is:

$$
\begin{cases}
m_{aj}\left(\ddot{x}_{aj} + \ddot{x}_c \cos\psi_{aj} + \ddot{y}_c \sin\psi_{aj}\right) - m_{aj}\left(2\omega_c \dot{y}_{aj} + \omega_c^2 x_{aj}\right) \\
\quad -F_{msaj}\sin\alpha + F_{mraj}\sin\alpha + F_{mabjx} + F_{bajx} = 0 \\
m_{aj}\left(\ddot{y}_{aj} - \ddot{x}_c \sin\psi_{aj} + \ddot{y}_c \cos\psi_{aj}\right) + m_{aj}\left(2\omega_c \dot{x}_{aj} - \omega_c^2 y_{aj}\right) F_{msaj}\cos\alpha \quad (j = 1,2,3) \\
\quad +F_{mraj}\cos\alpha + F_{mabjy} + F_{bajy} = 0 \\
J_{aj}\left(\ddot{\theta}_c + \ddot{\theta}_{aj}\right) + r_{baj} F_{msaj} - r_{baj} F_{mraj} + r_{baj} F_{mabj} = 0
\end{cases}
\tag{56}
$$

The bending–torsional coupling vibration differential equation of planetary carrier is:

$$
\begin{cases}
m_c \ddot{x}_c + \sum_{i=a,b}\sum_{j=1}^{3} m_{ij} \ddot{x}_c + \sum_{i=a,b}\sum_{j=1}^{3} m_{ij}\left(\ddot{x}_{ij}\cos\psi_{ij} - \ddot{y}_{ij}\sin\psi_{ij}\right) \\
\quad +\omega_c^2 \sum_{i=a,b}\sum_{j=1}^{3} m_{ij}\left(-x_{ij}\cos\psi_{ij} + y_{ij}\sin\psi_{ij}\right) + 2\omega_c \sum_{i=a,b}\sum_{j=1}^{3} m_{ij}\left(-\dot{x}_{ij}\sin\psi_{ij} - \dot{y}_{ij}\cos\psi_{ij}\right) \\
\quad -\sum_{j=1}^{3} F_{msajx} - \sum_{j=1}^{3} F_{mrajx} - \sum_{j=1}^{3} F_{mrbjx} + F_{bcx} = 0 \\
m_c \ddot{y}_c + \sum_{i=a,b}\sum_{j=1}^{3} m_{ij} \ddot{y}_c + \sum_{i=a,b}\sum_{j=1}^{3} m_{ij}\left(\ddot{x}_{ij}\sin\psi_{ij} + \ddot{y}_{ij}\cos\psi_{ij}\right) \\
\quad +\omega_c^2 \sum_{i=a,b}\sum_{j=1}^{3} m_{ij}\left(-x_{ij}\sin\psi_{ij} - y_{ij}\cos\psi_{ij}\right) + 2\omega_c \sum_{i=a,b}\sum_{j=1}^{3} m_{ij}\left(\dot{x}_{ij}\cos\psi_{ij} - \dot{y}_{ij}\sin\psi_{ij}\right) \\
\quad -\sum_{j=1}^{3} F_{msajy} - \sum_{j=1}^{3} F_{mrajy} - \sum_{j=1}^{3} F_{mrbjy} + F_{bcy} = 0 \\
J_c \ddot{\theta}_c + \sum_{i=a,b}\sum_{j=1}^{3} m_{ij} r_c^2 \ddot{\theta}_c + \sum_{j=1}^{3} J_{aj}\left(\ddot{\theta}_c + \ddot{\theta}_{aj}\right) - \sum_{j=1}^{3} J_{bj}\left(-\ddot{\theta}_c + \ddot{\theta}_{bj}\right) \\
\quad \pm \sum_{j=1}^{3} r_{bs} F_{msaj} - \sum_{j=1}^{3} r_{br1} F_{mraj} + \sum_{j=1}^{3} r_{br2} F_{mrbj} = -T_{out}
\end{cases}
\tag{57}
$$

The bending–torsional coupling vibration differential equation of the big gear ring is:

$$
\begin{cases}
m_{r2} \ddot{x}_{r2} + \sum_{j=1}^{3} F_{mrbjx} + F_{br2x} = 0 \\
m_{r2} \ddot{y}_{r2} - \sum_{j=1}^{3} F_{mrbjy} + F_{br2y} = 0 \\
J_{r2} \ddot{\theta}_{r2} - \sum_{j=1}^{3} r_{br2} F_{mrbj} = -T_{brake}
\end{cases}
\tag{58}
$$

The bending-torsional coupling vibration differential equation of short planetary wheel is:

$$
\begin{cases}
m_{bj}\left(\ddot{x}_{bj} + \ddot{x}_c \cos\psi_{bj} + \ddot{y}_c \sin\psi_{bj}\right) - m_{bj}\left(2\omega_c \dot{y}_{bj} + \omega_c^2 x_{bj}\right) \\
\quad +F_{mabjx} + F_{mrbj}\sin\alpha + F_{bbjx} = 0 \\
m_{bj}\left(\ddot{y}_{bj} - \ddot{x}_c \sin\psi_{bj} + \ddot{y}_c \cos\psi_{bj}\right) + m_{bj}\left(2\omega_c \dot{x}_{bj} - \omega_c^2 y_{bj}\right) \quad (j = 1,2,3) \\
\quad -F_{mabjy} + F_{mrbjy}\cos\alpha + F_{bbjy} = 0 \\
J_{bj}\left(-\ddot{\theta}_c + \ddot{\theta}_{bj}\right) - r_{bbj} F_{mabj} - r_{bbj} F_{mrbj} = 0
\end{cases}
\tag{59}
$$

In which $m_p$ is the equivalent mass of each component; $J_p$ is the equivalent moment of inertia of each inertial component; $p = s, c, a_j, b_j, r_1$ and $r_2$, respectively, represent each rotational component in the composite planetary gear; $T_{in}$ is the nonlinear excitation torque, considering the engine fluctuation torque $M_{in}$; $T_{out}$ is the load torque; $T_{brake}$ is the braking torque.

## 3. Dynamic Characteristic Analysis

### 3.1. The Inherent Characteristics

The modulus of all gears is 4 and the displacement coefficient is 0; the pressure angle is 20 rad; the meshing clearance is 0.06 mm; the base circular tooth thicknesses of planet gear a and planet gear b are 7.7592 mm and 7.8679 mm, which are used to calculate the phase difference between gears. The transmission ratio can be calculated according to the number of teeth of each gear; the main parameters as shown in Table 1.

**Table 1.** Main parameters of the transmission system.

|  | Number of Teeth | Moment of Inertia (kg*m$^2$) | Mass (kg) |
|---|---|---|---|
| The small ring r1 | 77 | 0.2546 | 11.275 |
| The big ring r2 | 82 | 0.4977 | 14.434 |
| sun gear | 34 | 0.0140 | 6.323 |
| planet gear a | 21 | 0.0028 | 2.163 |
| planet gear b | 22 | 0.0017 | 1.094 |

Based on the established dynamic model of the composite planetary gear system, the natural frequency and vibration mode of the system can be obtained. The analysis of its inherent characteristics can obtain the resonant frequency, vibration mode characteristics and vibration mode. The influence of the rotational inertia of each component of the system and the meshing stiffness between components on the natural frequency is further analyzed. The characteristics of the natural frequency and vibration mode of the composite planetary row under the working condition are analyzed, which can help to better understand the vibration law of the planetary gear system.

Figure 6 shows that the vibration modes of the composite planetary gear system can be divided into three types of modes: planetary gear vibration, global vibration and coupled vibration. When the system displays the planetary gear vibration, namely that only the planetary gear has torsional vibration, other rotating components do not vibrate, as shown in Figure 7a. When the system displays the global vibration, the vibration generated by the central rotating parts and planetary gear at all levels as well as the amplitude of the same type of planetary gear, are the same, as shown in Figure 7b. When the system displays the coupled vibration, except for the obvious characteristics of planetary wheel vibration and global vibration, the rest can be regarded as coupled vibration, as shown in Figure 7c.

### 3.2. The Natural Frequency Varies with the Trajectory of Stiffness and Inertia

The natural frequency is mainly related to the structural parameters of the system. It was found in the study that, with the change in parameters, the trajectory will appear as modal transition and trajectory intersection.

The trajectory of the natural frequencies of each order of the system changing with the meshing stiffness $k_{mraj}$ is shown in Figure 8. It can be seen that $\omega_2$ and $\omega_3$ are the double roots of the same frequency before point A, and their trajectories are the same. At point A, there is a modal transition, $\omega_2$ and $\omega_3$ alter rapidly with different frequencies, respectively, then $\omega_3$ changes with single frequency in the AB and $\omega_4$ changes with a single root frequency before point B. $\omega_3$ and $\omega_4$ modify in the same trajectory after the trajectory intersection at B until the mode transition occurs again at C, and $\omega_3$ and $\omega_4$ transform rapidly with different frequencies, respectively. In this case, the complex trajectory of natural frequencies is induced by the third-order natural frequency $\omega_3$. The trajectories of $\omega_5, \omega_6, \omega_8$ and $\omega_9$ always coincide, which is caused by the double roots under the same

natural frequency. Trajectories of the meshing stiffness $k_{msaj}$, $k_{mrbj}$ and $k_{mrbj}$ with natural frequency are similar to that of $k_{mraj}$, as shown in Figures 9–11.

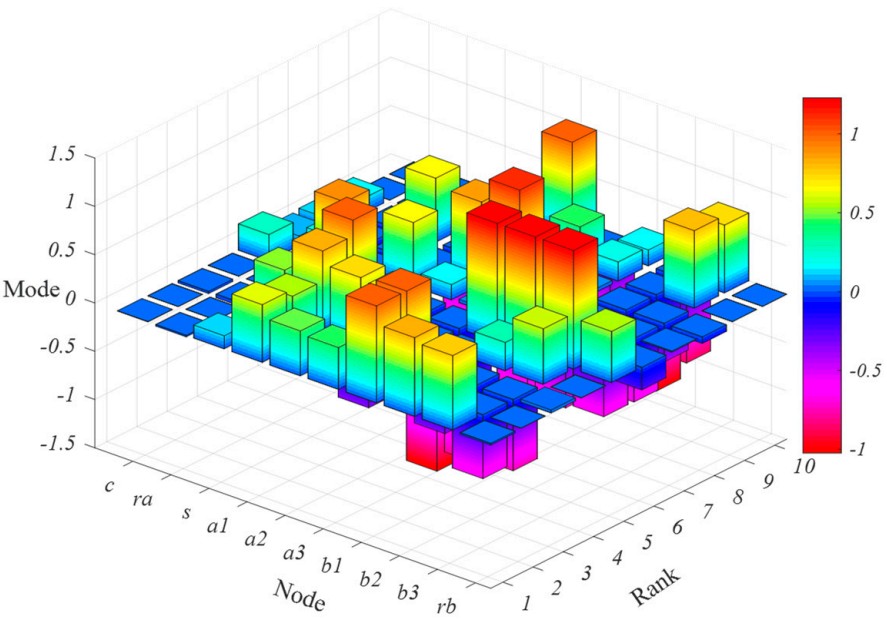

**Figure 6.** The vibration modes of each order of the composite planetary gear system.

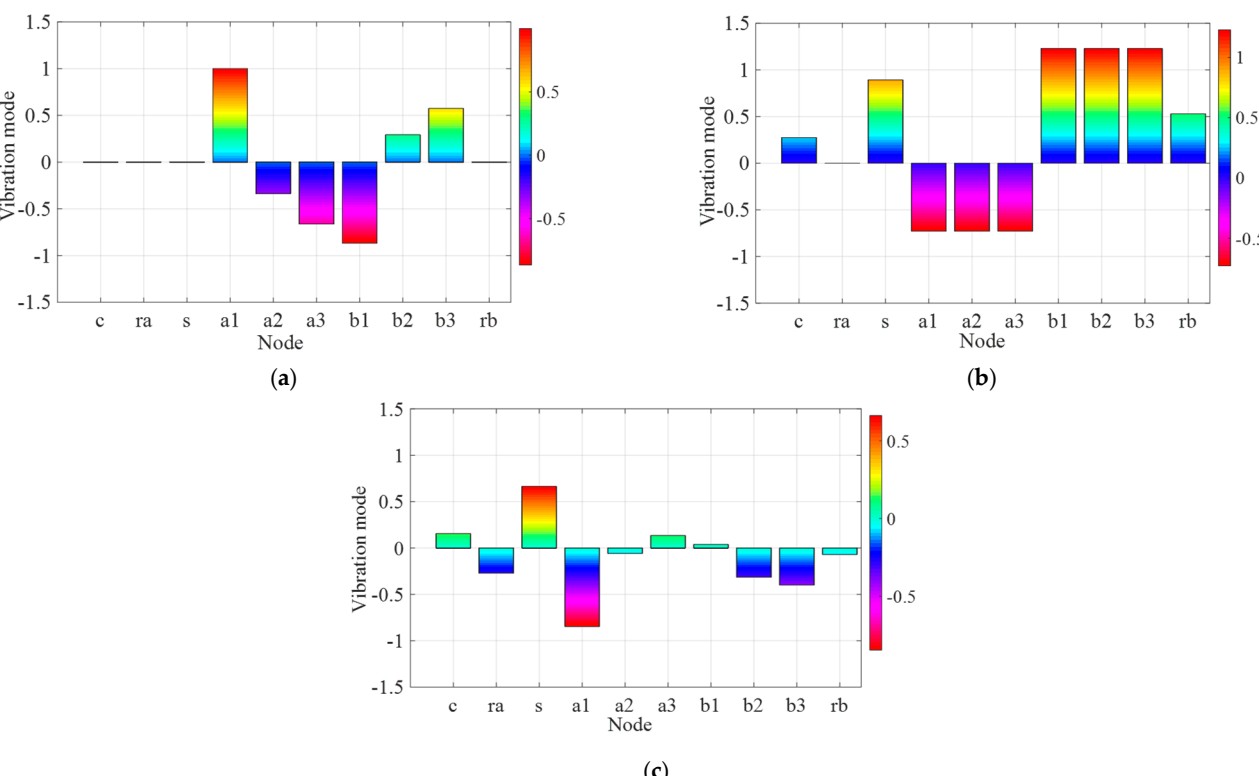

**Figure 7.** The vibration modes of each order of the composite planetary gear system: (**a**) Planetary wheel vibration; (**b**) Global vibration; (**c**) Coupled vibration.

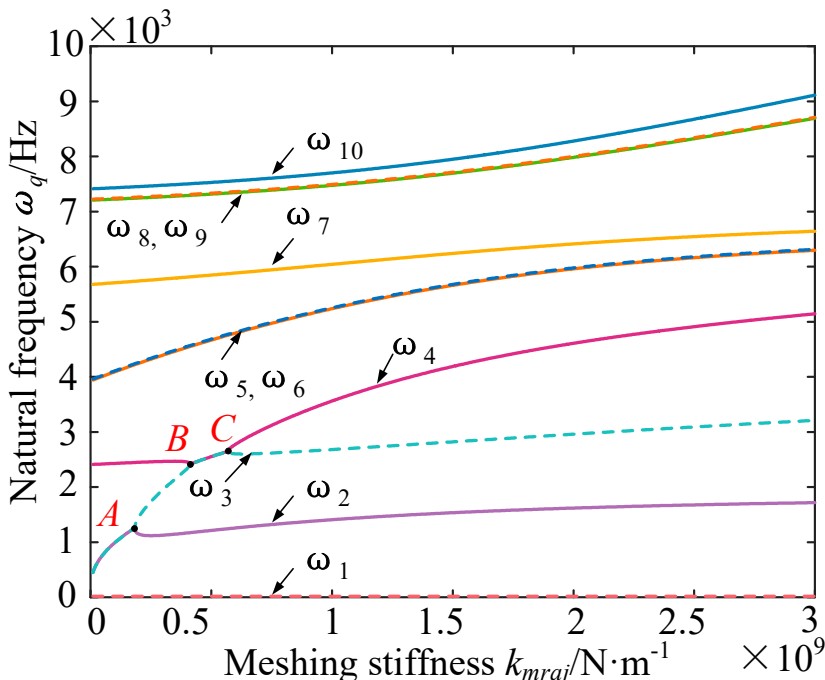

**Figure 8.** The trajectory of the natural frequency with $k_{mraj}$.

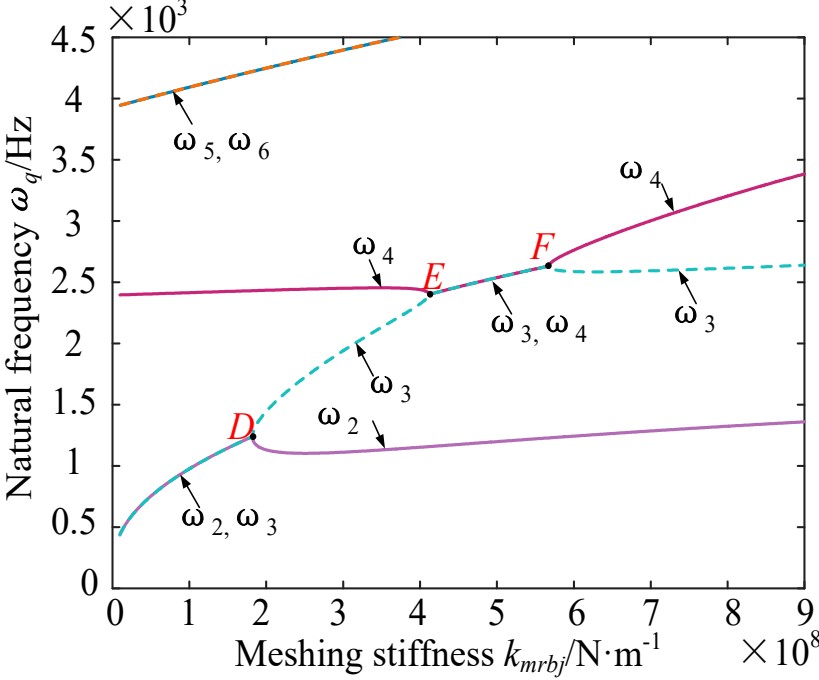

**Figure 9.** The trajectory of the natural frequency with $k_{msaj}$.

Shown in Figure 12 are the trajectory changes in natural frequencies of each order when the rotational inertia $J_{r1}$ of the gear ring increases from 0.01 kg·m² to 0.5 kg·m². It can be seen that there are more complex modal transitions and trajectory intersection phenomena when the natural frequency changes. The trajectory intersection of high-order frequencies $\omega_9$ and $\omega_{10}$ occur at point A, the two-order frequency trajectory remains consistent until the mode transition at point B separates rapidly; after that, these two phenomena occur simultaneously at point C. $\omega_7$ and $\omega_8$ change from the coincidence of trajectories before point C to the rapid separation, and then keep consistent with $\omega_9$, $\omega_6$ and $\omega_7$, which modify at a single frequency to point D and intersect their trajectories; until

the modal transition phenomenon occurs at point E, the two frequency trajectories quickly separate, and the same phenomenon as point C occurs again at point F. Before point F, the trajectories of $\omega_4$ and $\omega_5$ are the same. After point F, the trajectories of $\omega5$ and $\omega4$ are separated due to the modal transition and trajectory intersection, and $\omega_5$ and $\omega_6$ keep the same trajectory. This phenomenon is consistent with the torsional vibration, planetary wheel vibration and coupled vibration modes in the system. Trajectories of the inertia with natural $J_c$, $J_s$, $J_{r2}$, $J_{aj}$ and $J_{bj}$ frequency are similar to that of $J_{r1}$, as shown in Figures 13–17.

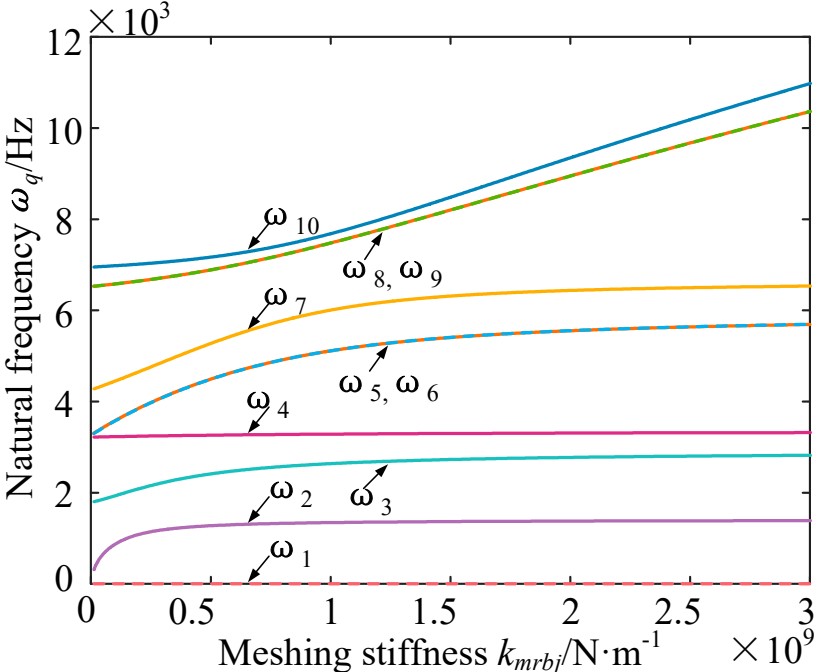

**Figure 10.** The trajectory of the natural frequency with $k_{mrbj}$.

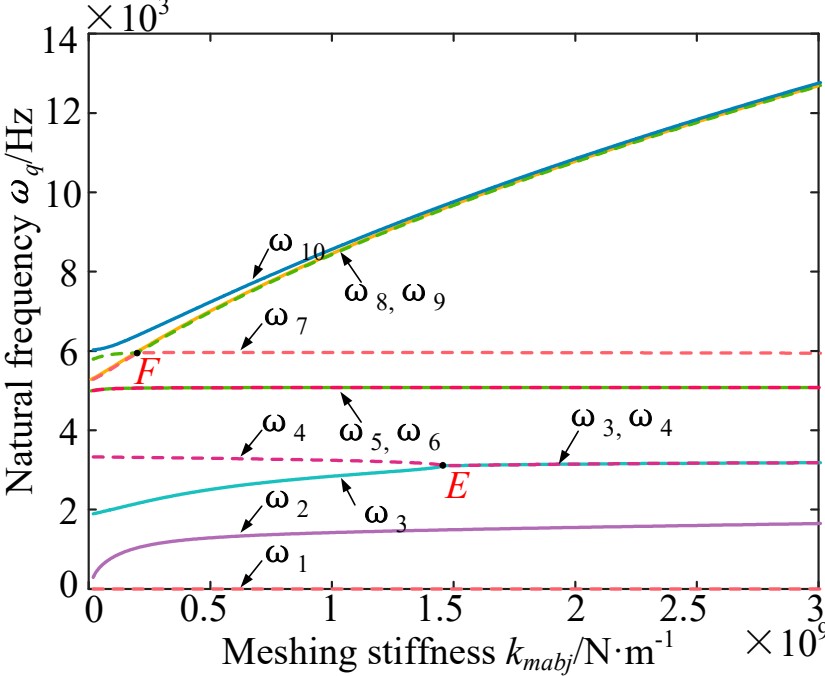

**Figure 11.** The trajectory of the natural frequency with $k_{mabj}$.

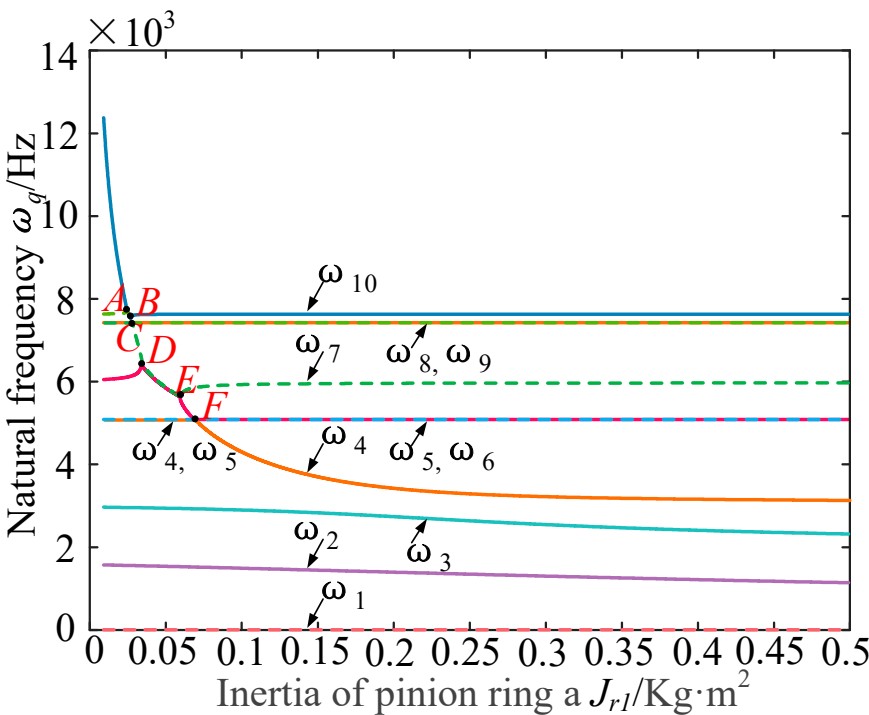

**Figure 12.** The trajectory of the natural frequency with $J_{r1}$.

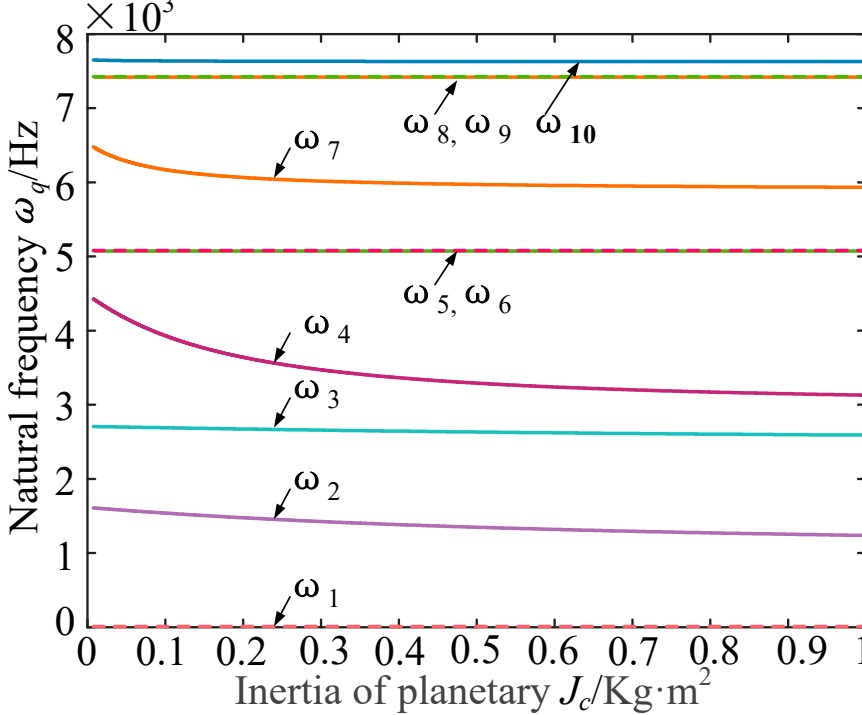

**Figure 13.** The trajectory of the natural frequency with $J_c$.

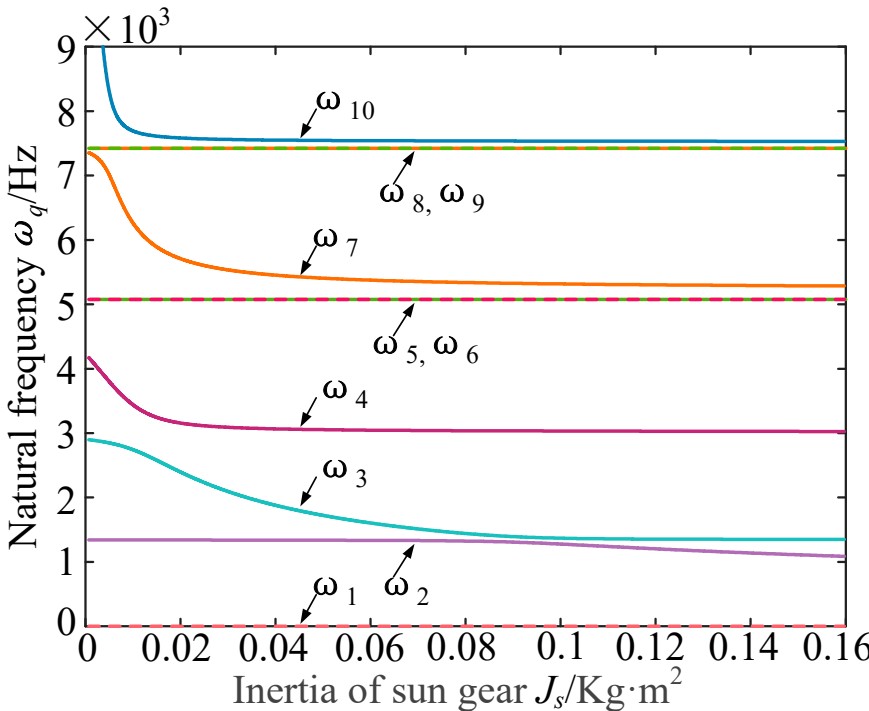

**Figure 14.** The trajectory of the natural frequency with $J_s$.

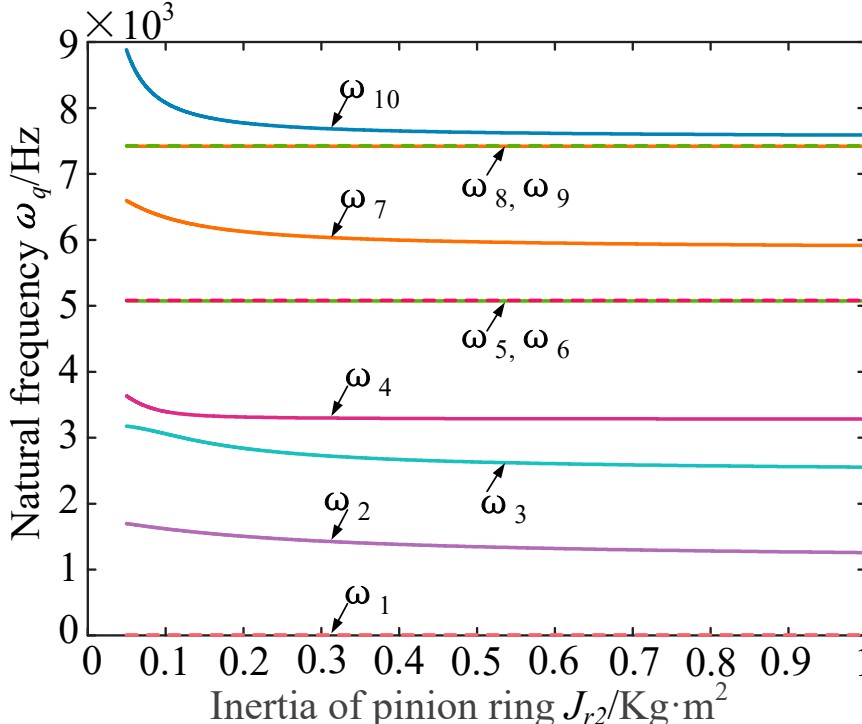

**Figure 15.** The trajectory of the natural frequency with $J_{r2}$.

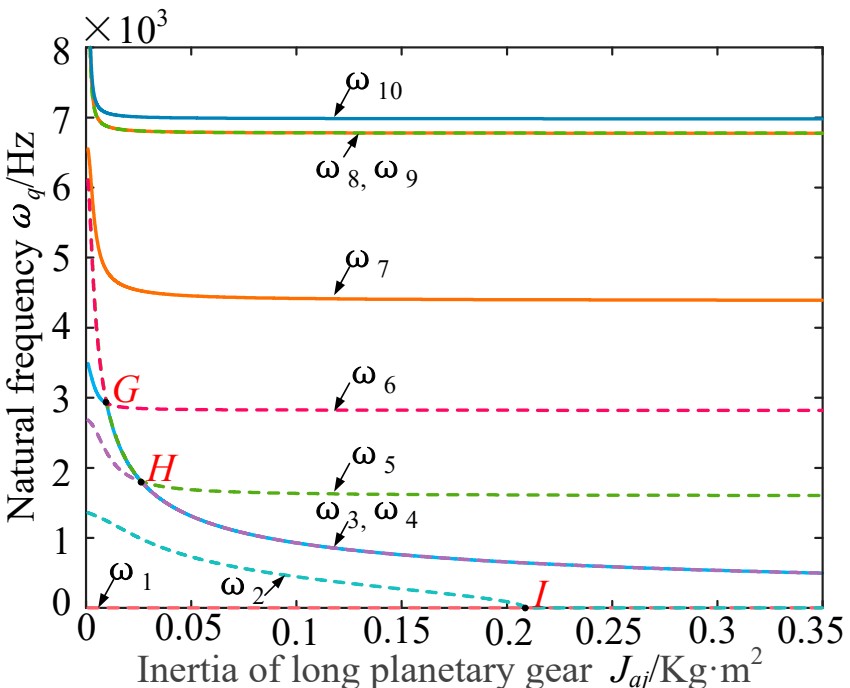

**Figure 16.** The trajectory of the natural frequency with $J_{aj}$.

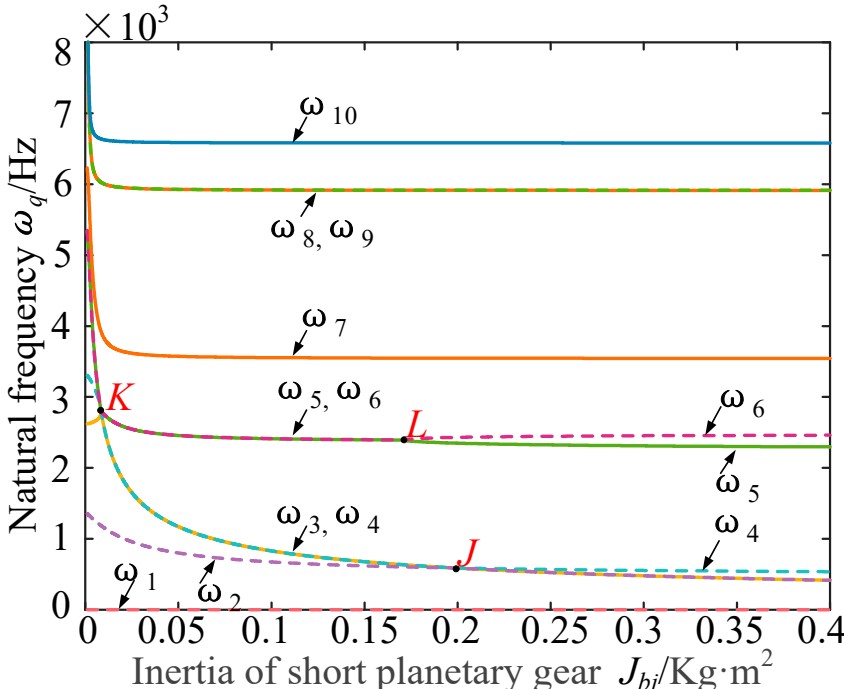

**Figure 17.** The trajectory of the natural frequency with $J_{bj}$.

### 3.3. Time-Frequency Characteristic Analysis of System Coupling Response

Simulation condition: the rated speed is 2300 r/min, the mean torque is 2000 N.m and the fluctuation frequency is six times the rotation frequency of the crankshaft.

As seen in Figure 18a,b, the planet carrier and the gear ring are completely overlapped in frequency; namely, their vibration is caused by the same excitation and their vibration frequency coincides completely. It can be seen that the low-frequency components of the vibration are mainly composed of engine rotation frequency $f_e$, modulation frequency $f_c$, $f_s-f_c$, the high-frequency components of the vibration are mainly composed of the planetary

gear rotary frequency $f_m$ and the engine rotation frequency $f_m \pm f_e$. It is also obvious that there are modulation components $f_m \pm 6f_e$ induced by the fluctuation frequency $6f_e$ and the meshing frequency $f_m$.

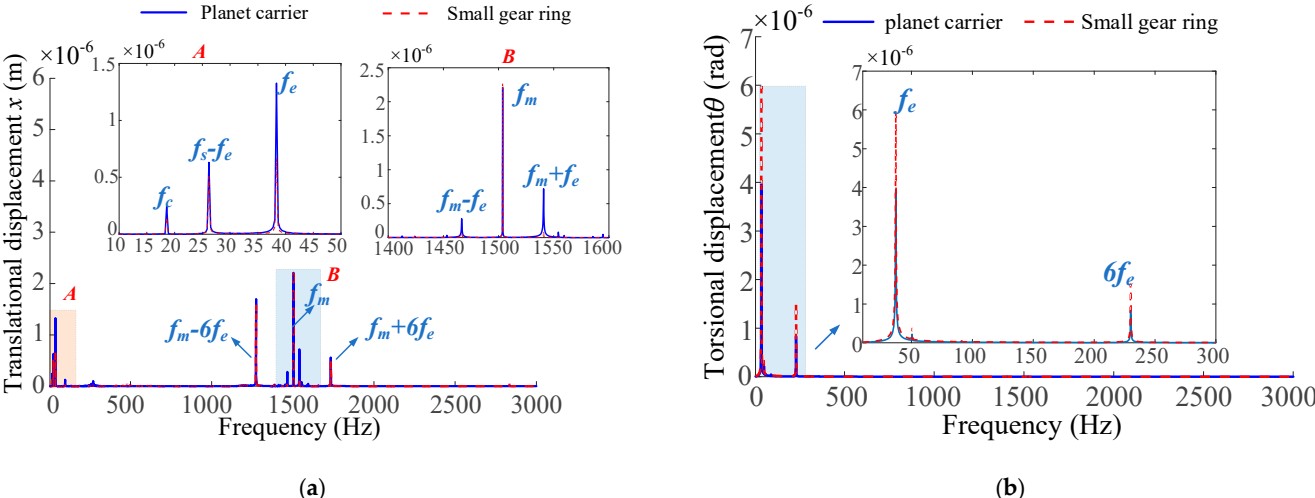

(a)                                       (b)

**Figure 18.** Frequency of vibration displacement of planetary carrier and small gear ring: (**a**) Spectrum of translational displacement; (**b**) Spectrum of torsional displacement.

It can be seen from Figure 19 that the main peak frequencies include the engine rotating frequency $f_e$, meshing frequency $f_m$ and its frequency doubling $nf_m$. In addition, there are sidebands with planetary carrier rotating frequency $f_e$, sun gear rotating frequency $f_s$, planetary wheel rotating frequencies $f_a$ and $f_b$, modulation frequencies $f_e$, $6f_e$, $f_b - f_e$ and $nf_m \pm f_c$, $nf_m \pm f_s$, $nf_m \pm f_a$ ($n$ = 1, 2, 3 … ). The largest amplitude is the meshing frequency and double frequency of the system, while the smallest amplitude is the rotational frequency of the system and modulation frequency with each meshing frequency. In addition, there are special frequencies, which are mainly caused by nonlinear factors. The dominant frequency of meshing force is the same from Figure 20, the maximum amplitude is $F_{mrb}$ and the smallest amplitude is $F_{msa}$.

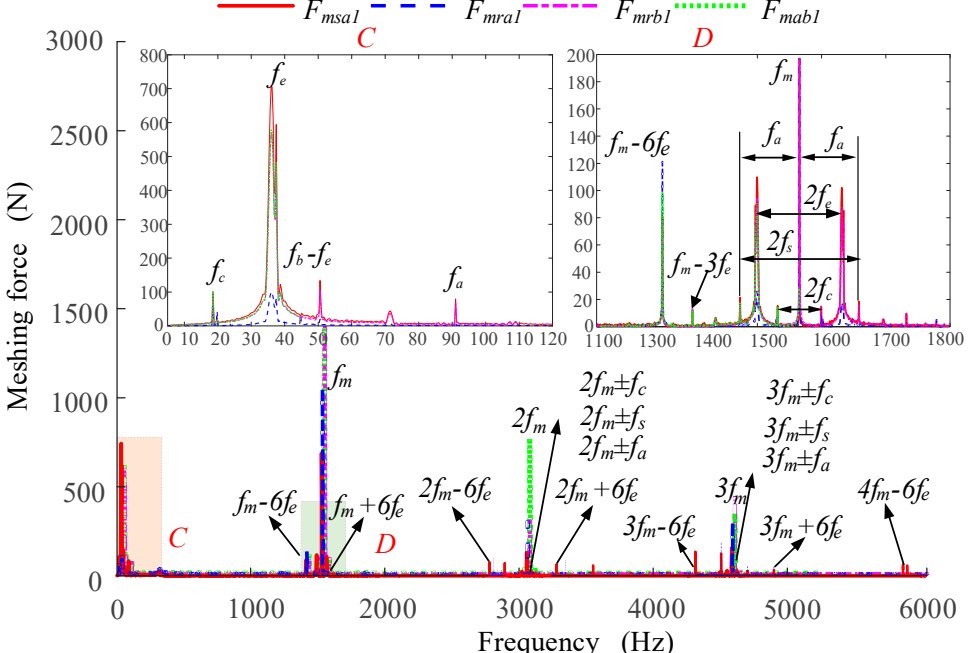

**Figure 19.** The spectrum of each meshing force.

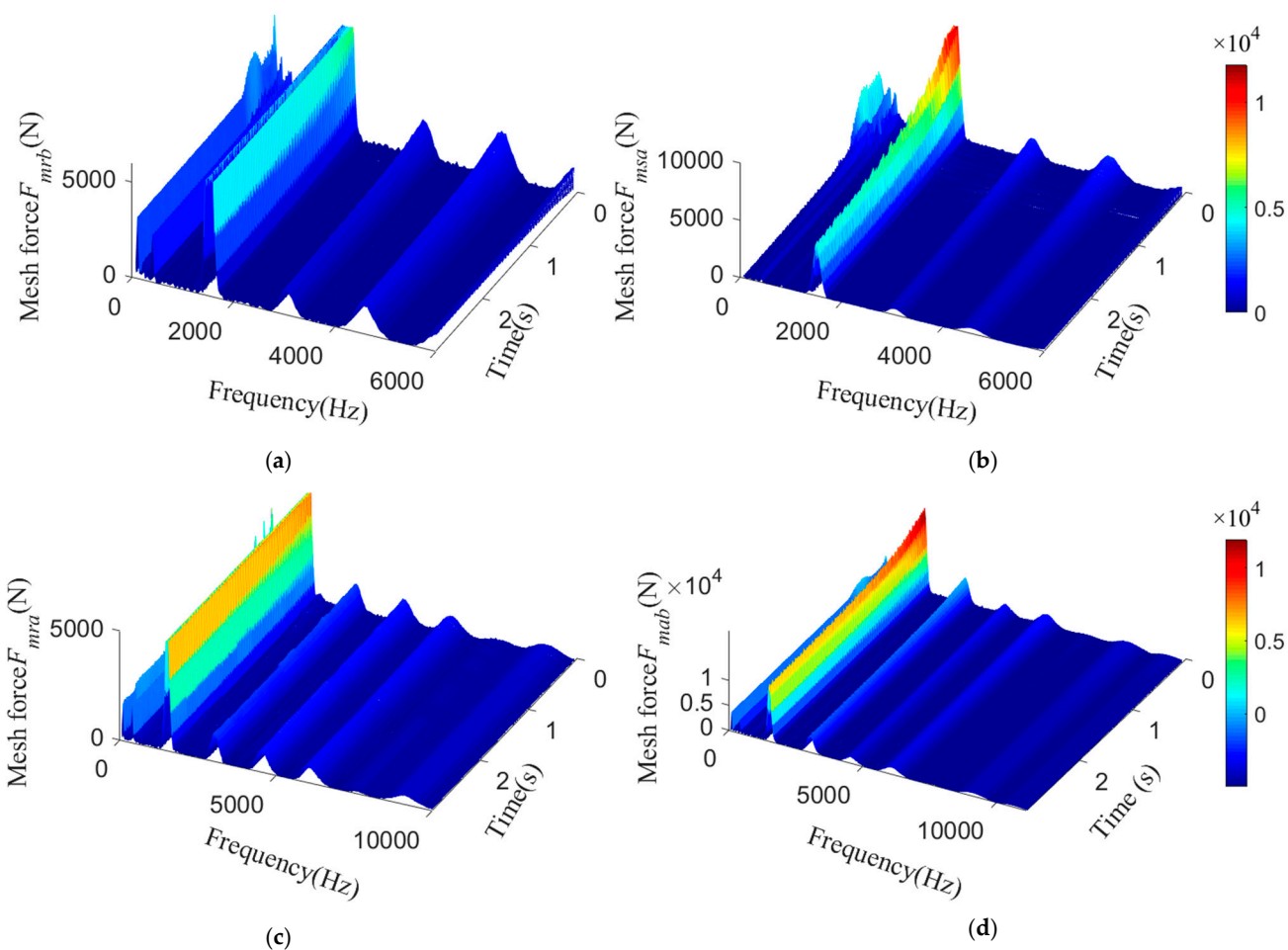

**Figure 20.** Time-frequency of each meshing force: (**a**) The meshing force $F_{mrb}$ between the large ring gear and planetary gear *b*; (**b**) The meshing force $F_{msa}$ between the sun gear and the planet gear; (**c**) The meshing force $F_{mra}$ between the small gear ring and the planetary gear *a*; (**d**) The meshing force $F_{mab}$ between planetary gears.

## 4. Verification of Model Accuracy

The virtual prototype model of the composite planetary gear system is established by combining CATIA and ADAMS for dynamic virtual simulation and vibration response analysis, the response results are compared and verified with the dynamic model.

### 4.1. Comparison of Natural Frequency

By comparing the mean value of the stabilized rotational speed with the calculated theoretical rotational speed, as shown in Table 2, it can be seen that the simulation errors of the rotational speed of the main components are all within 0.2%. It can be considered that the established virtual prototype model of the composite planetary wheel system has good accuracy and can be used for comparative analysis of numerical analysis results.

**Table 2.** Comparison between simulation value and theoretical values of rotational speed.

| | Ring Gear r1 (Input) | Sun Gear s | Planet Gear $a_j$ | Planet Gear $b_j$ | Planet Carrier c (Output) |
|---|---|---|---|---|---|
| Theoretical values (r/min) | 2300 | 1572.5 | 5463.1 | 3235.4 | 1113.8 |
| Simulation values (r/min) | 2300 | 1574.2 | 5452.7 | 3234.0 | 1113.9 |
| Errors (%) | 0 | 0.11 | 0.19 | 0.045 | 0.0048 |

The undamped natural frequency of the composite planetary gear system simulation model is obtained by modal analysis and the results are compared with the numerical results as shown in Table 3. Due to some assumptions of lumped mass method equivalence and inaccurate measurement of parameters such as inertia and stiffness being adopted in dynamic modeling, there is a certain error between simulation results and numerical calculation values; however, the overall error is relatively small.

**Table 3.** Comparison of natural frequency simulation value and theoretical value.

| Degree | 1 | 2 | 3 | 4 | 5 | 6 |
|---|---|---|---|---|---|---|
| Theoretical values(Hz) | 0 | 1339.1 | 2624.2 | 3281.3 | 5075.0 | 7420.5 |
| Simulation values (Hz) | 0 | 1324.9 | 2508.0 | 3206.6 | 5092.7 | 6970.2 |
| Errors(%) | 0 | 1.06 | 4.43 | 2.28 | 0.35 | 6.07 |

*4.2. Comparison of Meshing Force*

Simulation condition: the rated speed is 2300 r/min, the mean value is 2000 N.m and the fluctuation frequency is six times of the crankshaft frequency.

The numerical calculation values of meshing forces $F_{mra}$, $F_{mrb}$ and $F_{mab}$ are not completely consistent with the simulation results as shown in Table 4; the errors of all meshing forces are within 5%.

**Table 4.** Comparison between simulation and theoretical values of meshing forces.

| Mean Meshing Force | $F_{mra}$ | $F_{mrb}$ | $F_{mab}$ |
|---|---|---|---|
| Theoretical values (N) | 4360.4 | 6204.6 | 6177.3 |
| Simulation values (N) | 4391.9 | 5916.4 | 6295.9 |
| Errors (%) | 0.72 | 4.64 | 1.92 |

It can be clearly seen from Figures 21–23 that, in each meshing force spectrum, the vibration response of wave frequency $6f_e$ is dominant, followed by the system meshing frequency and double frequency $2f_m$ and $3f_m$, which are the same as that of numerical calculation. In the spectrum of the meshing force of the simulation results, there is a small value of the edge frequency band, mainly because the impact function is used in ADAMS to simulate the contact force of the gear meshing.

The comparative analysis of the virtual simulation and the numerical calculation results show that the coupling vibration response of the composite planetary gear system has a good correspondence in each vibration frequency component, which can form a good mutual verification with the established dynamic model and the numerical response results.

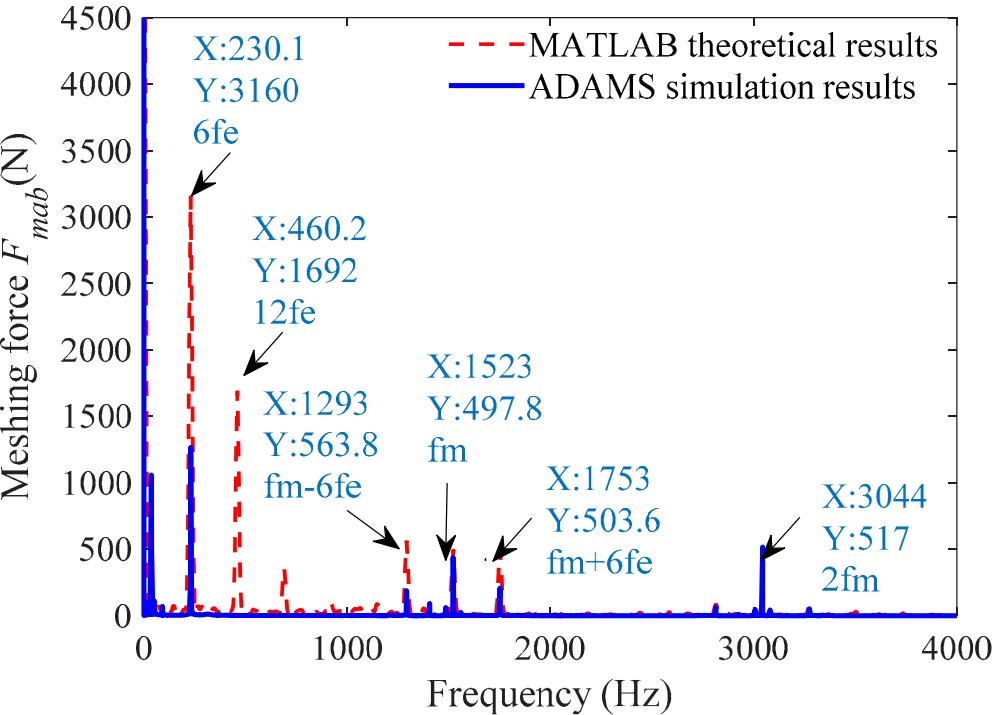

**Figure 21.** Frequency spectrum and time–frequency spectrum of meshing force $F_{mab}$.

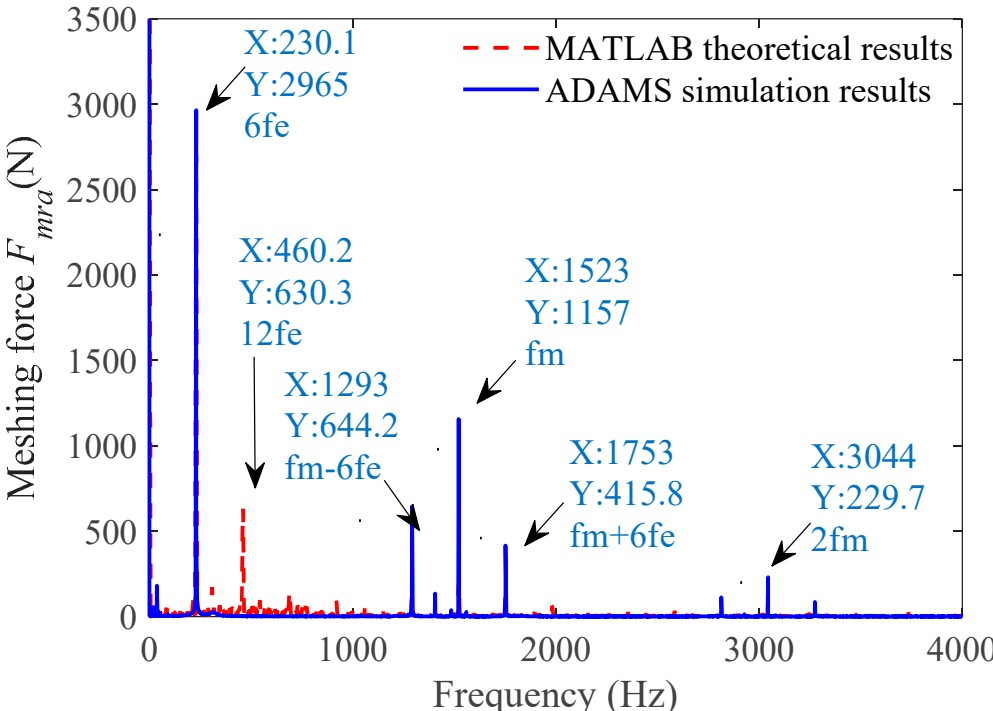

**Figure 22.** Frequency spectrum and time–frequency spectrum of meshing force $F_{mra}$.

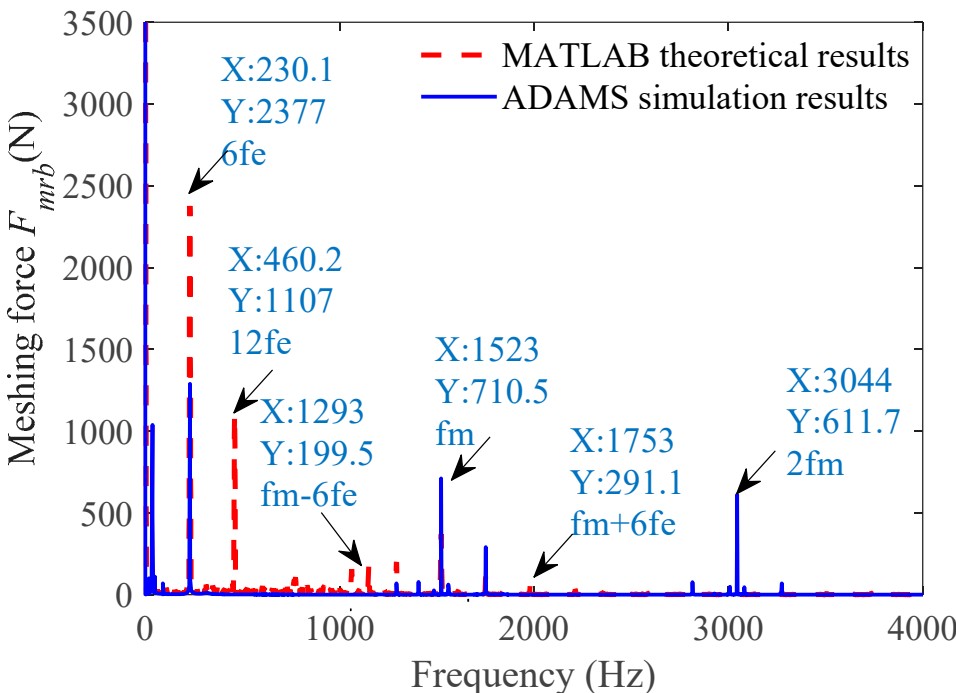

**Figure 23.** Frequency spectrum and time–frequency spectrum of meshing force $F_{mrb}$.

## 5. Conclusions

Taking time-varying stiffness, dynamic clearance, comprehensive error and engine harmonic excitation induced by the time-varying phase angle of the planetary gear into consideration, a time-varying dynamic model for composite planetary gear system is established in this paper. From this work, we can conclude that:

(1) The vibration of the composite planetary gear system has three modes: planetary gear vibration, global vibration and coupling vibration;

(2) The frequency trajectories of each order increase with the change in stiffness, although some frequency trajectories will have the phenomenon of modal transition and trajectory intersection. With the increase in inertia, the natural frequency of the system shows a decreasing trend and most orders change slightly. In addition, there are complex modal transitions and trajectory intersections in the natural frequency change trajectory, which are consistent with the torsional vibration, planetary wheel vibration and coupled vibration modes in the system;

(3) In bending–torsional direction, the vibration frequency components of different parts in the planetary gear are the same; however, their amplitudes are different. The low frequency is induced by the single frequency fluctuation torque converter input, which is independent of the system state. The high frequency is composed of the meshing frequency $fm$, six times the fluctuation frequency of engine rotating frequency $6fe$ and its modulation frequencies $fm \pm 6fe$. The rotation frequency and harmonic frequency of the engine have a great influence on the vibration response of the system.;

(4) The main frequencies in the meshing force spectrum include the engine rotation frequency and the meshing frequency and its frequency multiplication. In addition, there are sideband frequencies, such as the rotation frequencies of planetary carrier, sun gear and planetary gear as well as their modulation frequency. The larger amplitudes are the meshing frequency and its frequency doubling, while the smaller amplitudes are the rotation frequencies and the modulation frequency of meshing frequency; there are also some special frequencies, which are mainly caused by the nonlinear factors of the system.

**Author Contributions:** Conceptualization, T.C. and C.Z.; validation, Y.L.; writing—original draft preparation, T.C. and C.Z.; writing—review and editing, Y.L. and Y.C.; supervision, Y.L. All authors have read and agreed to the published version of the manuscript.

**Funding:** This research received no external funding.

**Conflicts of Interest:** The authors declare no conflict of interest.

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
