# Peer review of "Dynamic Modeling and Analysis of Nonlinear Compound Planetary System"

_machines, doi:10.3390/machines10010031_

Round 1

Reviewer 1 Report

(1)The part of introduction contains a lot of other scholars' work, but the main purpose is to illustrate three aspects of typical work, such as single-stage planetary row, nonlinear dynamic analysis and load sharing characteristics. It is recommended to pick typical notes rather than stack them.

(2)The "Input" of Figure 1 overlaps the model. The picture fails to describe the dynamic relationship between inputs and outputs.

(3)In Figure 2, multiple coordinate systems are used to describe the complex model, but the image labeling is not clear and the color differentiation is not obvious. It is suggested to adopt clearer map example.

(4)The formulas, for example from Eq. (11) to Eq. (16), are not aligned.

(5)The end of section 2.2.1 does not specify how to consider the impact of these errors on the system.

(6)The title of section 2.2.2 is "Internal Nonlinear Congestion", and the title of section 2.3 is "External Nonlinear Congestion". Titles of the same grade are not numbered in this way.

(7)The images from Figure 8 to Figure 11 should be formatted consistently for the same type of image.

(8)The formats of Figure 18 (a) and (b) are inconsistent. They should be consistent with other illustrations.

(9)The table fonts from Table 1 to Table 3 are inconsistent, and the upper and lower margins of the table are inconsistent

Author Response

Dear reviewer, the revised paper and response to the reviewer have been uploaded. Please see the attachment

Reviewer 2 Report

1. Supplement the design parameters of the transmission system, such as modulus, number of teeth, displacement coefficient, power, speed, transmission ratio, etc

2. The average meshing stiffness used in this paper is still time-varying meshing stiffness, so the detailed calculation method and process of meshing stiffness should be supplemented.

3. It is better to adopt time-varying meshing stiffness in this paper, otherwise the calculation accuracy will be greatly compromised

4. The mathematical model and equations of dynamics should be given in detail

5. Some references are inaccurate or need to be supplemented,such as reference[28]

 Analytical Investigation on Load Sharing Characteristics of Herringbone Planetary Gear Train with Flexible Support and Floating Sun Gear [J]. Mechanism and Machine Theory, 2020, 144 (2) : 1-27.

Author Response

Dear reviewer, the revised paper and the comments of the reviewer have been uploaded. Please see the attachment

Reviewer 3 Report

In the manuscript authors have studied the compound planetary gears and develop the bending-torsion coupling nonlinear dynamic model of the system based on Lagrange equation. The model is used to investigate the bending-torsion coupling meshing deformation relationship of each meshing pair along with the translational and torsion directions. The natural frequencies and vibration modal characteristics of the system are extracted from the model, and the influence of rotational inertia and meshing stiffness on the inherent characteristics of the system are studied. Moreover, the coupling vibration characteristics of the system under operating condition are analyzed in terms of the inherent characteristics and time- frequency characteristics of the system. The virtual prototype of the composite planetary system is used to verify the accuracy of the established model from speed, inherent characteristics, meshing force and frequency composition.

The manuscript is well written and well composed an suitable for publication in present form

Author Response

(The authors gave the same response as above.)

Round 2

Reviewer 1 Report

Thanks for the reply. I have no more questions.